# VloGraph: A Virtual Knowledge Graph Framework for Distributed Security Log Analysis

Kabul Kurniawan [1,2,*], Andreas Ekelhart [3,4], Elmar Kiesling [1], Dietmar Winkler [5], Gerald Quirchmayr [2] and A Min Tjoa [5]

1   Institute for Data, Process and Knowledge Management, Vienna University of Economics and Business, 1020 Vienna, Austria; elmar.kiesling@wu.ac.at
2   Research Group Multimedia Information Systems, University of Vienna, 1090 Vienna, Austria; gerald.quirchmayr@univie.ac.at
3   Research Group Security and Privacy, University of Vienna, 1090 Vienna, Austria; andreas.ekelhart@univie.ac.at
4   SBA Research, 1040 Vienna, Austria
5   Information and Software Engineering, Vienna University of Technology, 1040 Vienna, Austria; dietmar.winkler@tuwien.ac.at (D.W.); a.tjoa@tuwien.ac.at (A.M.T.)
*   Correspondence: kabul.kurniawan@wu.ac.at

**Abstract:** The integration of heterogeneous and weakly linked log data poses a major challenge in many log-analytic applications. Knowledge graphs (KGs) can facilitate such integration by providing a versatile representation that can interlink objects of interest and enrich log events with background knowledge. Furthermore, graph-pattern based query languages, such as SPARQL, can support rich log analyses by leveraging semantic relationships between objects in heterogeneous log streams. Constructing, materializing, and maintaining centralized log knowledge graphs, however, poses significant challenges. To tackle this issue, we propose VloGraph—a distributed and virtualized alternative to centralized log knowledge graph construction. The proposed approach does not involve any a priori parsing, aggregation, and processing of log data, but dynamically constructs a virtual log KG from heterogeneous raw log sources across multiple hosts. To explore the feasibility of this approach, we developed a prototype and demonstrate its applicability to three scenarios. Furthermore, we evaluate the approach in various experimental settings with multiple heterogeneous log sources and machines; the encouraging results from this evaluation suggest that the approach can enable efficient graph-based ad-hoc log analyses in federated settings.

**Keywords:** semantic log analysis; virtual log graphs; dynamic log extraction; decentralized logquerying; forensics





## 1. Introduction

Log data analysis is a crucial task in cybersecurity, e.g., when monitoring and auditing systems, collecting threat intelligence, conducting forensic investigations of incidents, and pro-actively hunting threats [1]. Currently available log analysis solutions, such as Security Information and Event Management (SIEM) systems, support the process by aggregating log data as well as storing and indexing log messages in a central relational database [2]. With their strict schemas, however, such databases are limited in their ability to represent links between entities [3]. This results in a lack of explicit links between heterogeneous log entries from dispersed log sources in turn makes it difficult to integrate the partial and isolated views on system states and activities reflected in the various logs. Furthermore, the central log aggregation model is also bandwidth-intensive and computationally demanding [2,4,5], which limits its applicability in large-scale infrastructures. Without a dedicated centralized log infrastructure, however, the process necessary to acquire, integrate and query log data are tedious and inefficient, which poses a key challenge for security analysts who often face time critical tasks.

To illustrate the issue, consider the example in Figure 1. It is based on log data produced by multi-step attacks as described in [6]. These log data sets will also be used in a scenario in Section 7. The various steps of the attack are reflected in a large number of log messages in a diverse set of log sources dispersed across multiple hosts and files (e.g., *Syslog, ApacheLog, AuthLog, MailLog* etc.). *Vulnerability Scan*, for instance—which scans a system for known vulnerabilities—leaves some traces in multiple log sources such as *Syslog* and *ApacheLog* on *Host1* and *Host3*, respectively. *User Enumeration*—an activity that aims to guess or confirm valid users in a system—also leaves some traces in (*AuthLog, MailLog etc.*) stored on *Host1* and *Host2*. As this example shows, a single attack step typically results in a large number of log events that capture comprehensive information. This information can be used for log analysis and attack investigation, but correlating, tracing, and connecting the individual indicators of compromise—e.g., through timestamps, IP addresses, user names, processes and so forth—is typically a challenging and often time-consuming task. This is partly due to the weak structure of log sources and their inconsistent format and terminologies. Consequently, it is difficult to get a complete picture of suspicious activities and understand what happened in a given attack—particularly in the face of fast evolving, large volume, and highly scattered log data.

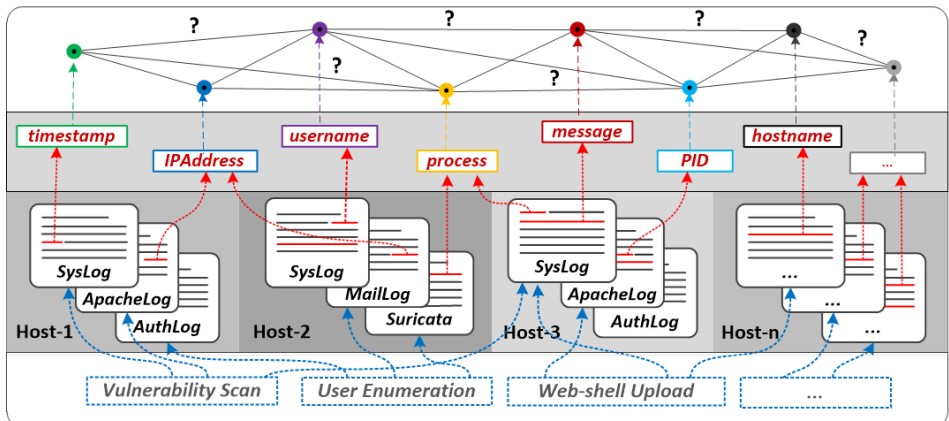

**Figure 1.** Motivating example illustrating that attack steps leave traces in various log sources across multiple hosts, making it difficult to reconstruct what happened.

To tackle these challenges, we propose *VloGraph*, a decentralized framework to contextualize, link, and query log data. We originally introduced this framework in [7]; in this paper, we extend this prior work with a detailed requirements specification, evaluation with two additional application scenarios, and a section reflecting upon graph-based log integration and analysis, decentralization and virtualization, and discussing applications and limitations.

More specifically, we introduce a method to execute federated, graph pattern-based queries over dispersed, heterogeneous raw log data by dynamically constructing virtual knowledge graphs [8,9]. This knowledge-based approach is designed to be decentralized, flexible and scalable. To this end, it (i) federates graph-pattern based queries across endpoints, (ii) extracts only potentially relevant log messages, (iii) integrates the dispersed log events into a common graph, and (iv) links them to background knowledge.

All of these steps are executed at query time without any up-front ingestion and conversion of log messages.

Figure 2 illustrates the proposed approach; the virtual log knowledge graph at the center of the figure is constructed dynamically from dispersed log sources based on analysts' queries and linked to external and internal knowledge sources.

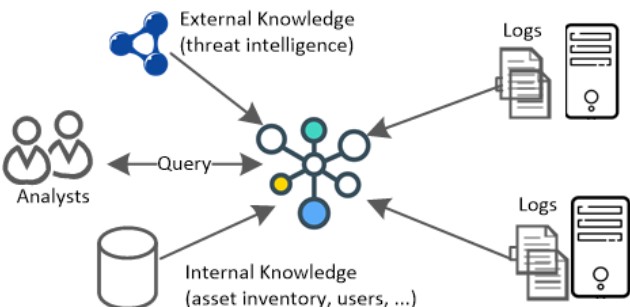

**Figure 2.** Concept overview.

A key advantage of the graph-based model of this virtual knowledge graph is that it provides a concise, flexible, and intuitive abstraction for the representation of various relations such as, e.g., connections in networked systems, hierarchies of processes on endpoints, associations between users and services, and chains of indicators of compromise. These connections automatically link log messages that are related through common entities (such as users, hosts, and IP addresses); such links are crucial in cybersecurity investigations, as threat agent activities typically leave traces in various log files that are often spread across multiple endpoints in a network, particularly in discovery, lateral movement, and exfiltration stages of an attack ATT&CK Matrix for Enterprise [10].

In contrast to traditional workflows that store log messages in a centralized repository, *VloGraph* shifts the log parsing workload from ingestion to analysis time. This makes it possible to directly access and dynamically integrate the most granular raw log data without any loss of information that would occur if the logs were pre-filtered and aggregated—typical steps performed before transferring them to a central archive.

*VloGraph* tackles a number of pressing challenges in security log analysis (discussed in Section 4) and facilitates (i) ad-hoc integration and semantic analyses on raw log data without prior centralized materialization, (ii) the collection of evidence-based knowledge from heterogeneous log sources, (iii) automated linking of fragmented knowledge about system states and activities, and (iv) automated linking to external security knowledge (such as, e.g., attack patterns, threat implications, actionable advice).

The remainder of this paper is organized as follows: Section 2 introduces background knowledge as conceptual foundation, including semantic standards and virtual knowledge graphs. Section 3 provides an overview of related work in this area, and in Section 4, we discuss challenges in log analysis and derive requirements for our approach. Section 5 introduces the proposed *VloGraph* architecture and describes the components for virtual log knowledge graph construction in detail. In Section 6 we present a prototypical implementation of the architecture and illustrate its use in three application scenarios. We evaluate our approach on a systematically generated log dataset in Section 7 and discuss benefits and limitations of the presented approach in Section 8. Finally, we conclude with an outlook on future work in Section 9.

## 2. Background

In this section, we first provide a brief background on log files and formats and then introduce knowledge graphs as a conceptual foundation of our approach.

### Log File Formats

Typically, software systems (operating systems, applications, network devices, etc.) produce time-sequenced log files to keep track of relevant events. These logs are used by roles such as administrators, security analysts, and software developers to identify and diagnose issues. Various logging standards are in use today, often focused on a specific application domain, such as operating system logs (e.g., syslogd [11] and Windows Event Logs [12]), web server logs (e.g., W3C Extended log file format [13], NGINX logging [14]), database logs, firewall logs, etc.

Log entries are often stored as semi-structured lines of text, comprising structured parts (e.g., timestamp and severity level) and unstructured fields such as a message. While the structured parts are typically standardized, the content of the unstructured fields contains context specific information and lacks uniformity. Before plain text log lines can be (automatically) analyzed, they must be split into their relevant parts, e.g., into a key-value based representation. This pre-processing step often relies on predefined regular expressions. Other standards, such as the Windows Event Log (EVTX), are already highly structured and support XML or JSON. Despite standardization attempts, heterogeneous log formats are still often an impediment to effective analysis. Current research also strives to automatically detect structure in log files [15] and to establish a semantic understanding of log contents such as [16,17].

### Knowledge Graphs

A knowledge graph is a directed, edge-labelled graph $G = (V, E)$ where $V$ is a set of vertices (nodes) and $E$ is a set of edges (properties). A single graph is usually represented as a collection of triples $T = < s\ p\ o >$ where $s$ is a *subject*, $p$ is a *predicate*, and $o$ is an *object*.

### RDF, RDF-S, OWL

Resource Description Framework (RDF) is a standardized data model that has been recommended by the W3C [18] to represent directed edge-labelled graphs. In RDF, a *subject* is a resource identified by a unique identifier (URI) or a blank-node, an *object* can be a resource, blank-node or literal (e.g., String, number), and *predicate* is a property defined in an ontology and must be a URI.

Figure 3 shows an excerpt of a log knowledge graph that expresses a single Apache log event in RDF. To the left, it shows a graphical representation of this log event and to the right a representation of the same graph in TURTLE [19] serialization. The subject *:logEntry-24e* is characterized by a number of properties that specify its type (*cl:ApacheLog*), the timestamp of creation, the originate host of the log event, the client that made the request, and the request string. Furthermore, the highlighted IP-Address (in the visualization) indicate that the objects link to other entities in the graph.

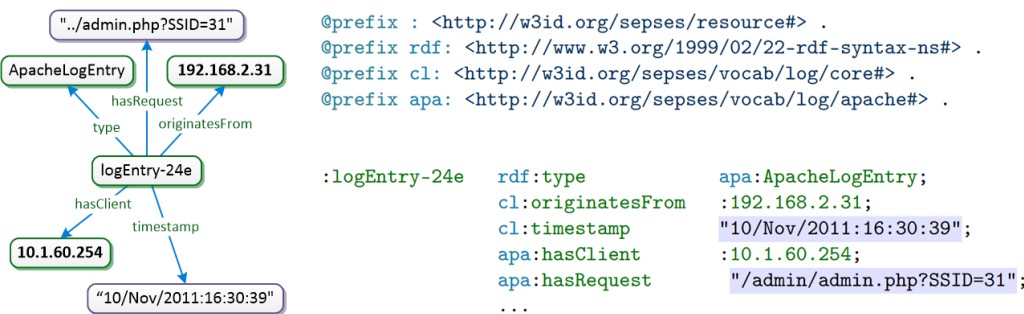

**Figure 3.** Excerpt of an RDF log graph generated from an Apache log event.

RDF-S (Resource Description Framework Schema) [20] is a W3C standard data model for knowledge representation. It extends the basic RDF vocabulary with a set of classes and RDFS entailment (inference patterns) [21]. OWL (Ontology Web Language) [22] is also a W3C standard for authoring ontologies.

### SPARQL

SPARQL [23] is a W3C query language to retrieve and manipulate data stored in RDF. It offers rich expressivity for complex queries such as aggregation, subqueries, and negation. Furthermore, SPARQL provides capabilities to express queries across multiple distributed data sources through SPARQL query federation [24]. In the security context, this is a major benefit, as security-relevant information is typically dispersed across different systems and networks and requires the consideration of e.g., different log sources, IT repositories, and cyberthreat intelligence sources [25].

**Virtual Knowledge Graphs**

The Virtual Knowledge Graph (VKG) paradigm for data integration is typically used to provide integrated access to heterogeneous relational data. The approach—also known in the literature as Ontology-based Data Access—aims to replace the rigid structure of tables in traditional data integration layers with the flexibility of graphs. This makes it possible to connect data silos by means of conceptual graph representations that provide an integrated view on the data. To this end, VKGs integrate three main ideas [9]:

- *Data Virtualization (V)*, i.e., they provide a conceptual view that avoids exposing end users to the actual data sources. This conceptual view is typically not materialized in order to make it possible to query the data without paying a price in terms of storage and time for the data to be made accessible.
- *Domain Knowledge (K)*, i.e., the graphs can be enriched and contextualized with domain knowledge that makes it possible to derive new implicit knowledge from the asserted facts at the time a query is executed.
- *Graph Representation (G)*, i.e., the data are represented as graph where object and data values are represented as nodes, and properties of those object are represented as edges. Compared to traditional relational integration tables, the graph representation provides flexibility and through mapping and merging makes it easier to link and integrate data.

In this paper, we follow these principles and generalize the VKG paradigm to weakly structured log data.

### 3. Related Work

In this section, we organize the related literature—from general to specific—into three categories:

**Log Management and Analytics**

The rising number, volume and variety of logs has created the need for systematic computer security log management [26] and motivated the development of a wide range of log-analytic techniques to derive knowledge from these logs [27], including anomaly detection [28,29], clustering [30], and rule-base intrusion detection [31].

In the context of our work, approaches that aim to integrate and analyze log data across multiple sources are particularly relevant. Security Information and Event Management (SIEM) are widely used to provide a centralized view on security-relevant events inside an organization and focus on data aggregation, correlation, and typically rule-based alerting. These ideas are outlined in numerous guidelines and industrial best practices such as the NIST Cybersecurity Framework [32] and NIST SP 800-92 Guide to Computer Security Log Management [33]. In this current state of practice, various commercial offerings provide centralized solutions e.g. Gartner Magic Quadrant for SIEM 2021 [34].

Whereas SIEMs facilitate centralized log aggregation and management, however, they lack a semantic foundation for the managed log data and consequently typically do not make it easy to link, contextualize, and interpret events against the background of domain knowledge. To tackle these challenges, Ref. [35] creates a foundation for semantic SIEMs that introduces a Security Strategy Meta-Model to enable interrelating information from different domains and abstraction levels. In a similar vein, Ref. [2] proposes a hybrid relational-ontological architecture to overcome cross-domain modeling, schema complexity, and scalability limitations in SIEMs. This approach combines existing relational SIEM data repositories with external vulnerability information, i.e., Common Vulnerabilities and Exposures (CVE) [36].

**Graph-Based Log Integration and Analysis**

More closely related to the *VloGraph* approach proposed in this paper, a stream of literature has emerged that recognizes the interrelated nature of log data and conceives log

events and their connections as graphs—i.e., labeled property graphs (LPGs) or semantically explicit RDF knowledge graphs.

In the former category, LPGs are stored in graph databases and queried through specialized graph query languages. For network log files, for instance, Ref. [37] proposes an approach that materializes the log in a Neo4J graph database and makes it available for querying and visualization. The approach is limited to a single log source and focuses exclusively on network log analysis. Similar to this, CyGraph [38] is a framework that integrates isolated data and events in a unified graph-based cybersecurity model to assist decision making and improve situational awareness. It is based on a domain-specific language CyQL to express graph patterns and uses a third-party tool for visualization.

Another stream of literature transforms logs into RDF knowledge graphs that can be queried with SPARQL, a standardized query language. Early work such as [39] has illustrated that the use of explicit semantics can help to avoid ambiguity, impose meaning on raw log data, and facilitate correlation in order to lower the barrier for log interpretation and analysis. In this case, however, the log source considered is limited to a firewall log. Approaches like this do not directly transform log data into a graph, but impose semantics to existing raw log data or log data stored in a relational database. More recently, approaches have been developed that aim to transform log data from multiple sources into an integrated log knowledge graph.

For structured log files, Ref. [40] discusses an approach that analyzes their schema to generate a semantic representation of their contents in RDF. Similar to our work, the approach links log entities to external background knowledge (e.g., DBPedia), but the log source processed is limited to a single log type. Ref. [41] leverages an ontology to correlate alerts from multiple Intrusion Detection Systems (IDSs) with the goal of reducing the number of false-positive and false-negative alerts. It relies on a shared vocabulary to facilitate security information exchange (e.g., IDMEF, STIX, TAXII), but does not facilitate linking to other log sources that may contain indicators of attacks.

LEKG [42] provides a log extraction approach to construct knowledge graphs using inference rules and validates them from a background knowledge graph. It uses local inference rules to create graph elements (triples) which can later be used to identify and generate causal relations between events. Compared to *VloGraph*, the approach does not aim to provide integration and interlinking over multiple heterogeneous log sources.

To facilitate log integration, contextualization and linking to background knowledge, Ref. [17] proposes a modular log vocabulary that enables log harmonization and integration between heterogeneous log sources. A recent approach proposed in [25] introduces a vocabulary and architecture to collect, extract, and correlate heterogeneous low-level file access events from Linux and Windows event logs.

Compared to the approach in this paper, the approaches discussed so far rely on a centralized repository. A methodologically similar approach for log analysis outside of the security domain has also been introduced in [43], which leverages ontology-based data access to support log extraction and data preparation on legacy information systems for process mining. In contrast to this paper, the focus is on log data from legacy systems in existing relational schemas and on ontology-based query translation.

**Decentralized Security Log Analysis**

Decentralized event correlation for intrusion detection was introduced in early work such as [44], where the authors propose a specification language to describe intrusions in a distributed pattern and use a peer-to-peer system to detect attacks. In this decentralized approach, the focus is on individual IDS events only. To address scalability limitations of centralized log processing, Ref. [4] distributes correlation workloads across networks to the event-producing hosts. Similar to this approach, we aim to tackle challenges of centralized log analysis. However, we leverage semantic web technologies to also provide contextualization and linking to external background knowledge. In the cloud environment, Ref. [45] proposes a distributed and parallel security log analysis framework that provides analyses of a massive number of systems, networks, and transaction logs in a scalable

manner. It utilizes the two-level master-slave model to distribute, execute, and harvest tasks for log analysis. The framework is specific to cloud-based infrastructures and lacks the graph-oriented data model and contextualization and querying capabilities of our approach.

## 4. Requirements

Existing log management systems typically ingest log sources from multiple log-producing endpoints and store them in a central repository for further processing. Before they can be analyzed, such systems typically parse and index these logs, which typically requires considerable amounts of disk space to store the data as well as computational power for log analysis. The concentrated network bandwidth, CPU, memory, and disk space needs limit the scalability of such centralized approaches.

Decentralized log analysis, by contrast, (partly) shifts the computational workloads involved in log pre-processing (e.g., acquisition, extraction, and parsing) and analysis to the log-producing hosts [4]. This model has the potential for higher scalability and applicability in large-scale settings where the scope of the infrastructure prohibits effective centralization of all potentially relevant log sources in a single repository.

Existing approaches for decentralized log processing, however, primarily aim to provide correlation and alerting capabilities, rather than the ability to query dispersed log data in a decentralized manner. Furthermore, they lack effective means for semantic integration, contextualization, and linking, i.e., dynamically creating connections between entities and potentially involving externally available security information. They also typically have to ingest all log data continuously on the local endpoints, which increases continuous resource consumption across the infrastructure.

In this paper, we tackle these challenges and propose a distributed approach for security log integration and analysis. Thereby, we facilitate ad-hoc querying of dispersed raw log sources without prior ingestion and aggregation in order to address the following requirements (*R*):

- *R.1—Resource-efficiency*   Traditional log management systems, such as SIEMs, perform continuous log ingestion and preprocessing, typically from multiple monitoring endpoints, before analyzing the log data. This requires considerable resources as all data needs to be extracted and parsed in advance. A key requirement for distributed security log analysis is to avoid unnecessary ex-ante log preprocessing (acquisition, extraction, and parsing), thus minimizing resource requirements in terms of centralized storage space and network bandwidth. This should make log analysis both more efficient and more scalable.

- *R.2—Aggregation and integration over multiple endpoints*   As discussed in the context of the motivating example in Section 1, a single attack may leave traces in multiple log sources, which can be scattered across different systems and hosts. To detect sophisticated attacks, it is therefore necessary to identify and connect such isolated indicators of compromise [17]. The proposed solution should therefore provide the ability to execute federated queries across multiple monitoring endpoints concurrently and deliver integrated results. This makes it possible to detect not only potential attack actions, but also to obtain an integrated picture of the overall attack (e.g., through linking of log entries).

- *R.3—Integration, Contextualization & Background-Linking*   the interpretation of log information for attack investigation depends highly on the context; isolated indicators on their own are, however, often inconspicuous in their local context. Therefore, the proposed approach should provide the ability to contextualize disparate log information, integrate it, and link it to internal background knowledge and external security information.

- *R.4—Standards-based query language*   The proposed approach should provide an expressive, standards-based query language for log analysis. This should make it easier for analysts to formulate queries (e.g., define rules) during attack investigation in an intuitive and declarative manner.

## 5. VloGraph Framework Architecture

Based on the requirements set out in Section 4, we propose *VloGraph*, an approach and architecture for security log analytics based on the concept of Virtual Knowledge Graphs (VKGs). The proposed approach leverages Semantic Web Technologies that provide (i) a standardized graph-based representation to describe data and their relationships flexibly using RDF [46], (ii) semantic linking and alignment to integrate multiple heterogeneous log data and other resources (e.g., internal/external background knowledge), and (iii) a standardized semantic query language (i.e., SPARQL [23]) to retrieve and manipulate RDF data.

To address R.1, our approach does not rely on centralized log processing, i.e., we only extract relevant log events based on the temporal scope and structure of a given query and its query parameters. Specifically, we only extract lines in a log file that: (i) are within the temporal scope of the query, and (ii) may contain relevant information based on the specified query parameters and filters.

The identified log lines are extracted, parsed, lifted to RDF, compressed, and temporarily stored in a local cache on the respective endpoint. This approach implements the concept of data virtualization and facilitates on-demand log processing. By shifting computational loads to individual monitoring agents and only extracting log entries that are relevant for a given query, this approach can significantly reduce unnecessary log data processing. Furthermore, due to the use of RDF compression techniques, the transferred data are rather small; we discuss this further in Section 7.

To address R.2, we distribute queries over multiple log sources across distributed endpoints and combine the results in a single integrated output via query federation [24].

To address R.3, we interlink and contextualize our extracted log data with internal and external background knowledge—such as, e.g., IT asset information and cybersecurity knowledge—via semantic linking and alignment. Finally, we use SPARQL to formulate queries and perform log analyses, which addresses R.4. We will illustrate SPARQL query federation and contextualization in multiple application scenarios for in Section 6.

Figure 4 illustrates the *VloGraph* virtual log graph and query federation architecture for log analysis; (i) a **Log Parser** on each host, which receives and translates queries, extracts raw log data from hosts, parses the extracted log data into an RDF representation, compresses the resulting RDF data into a binary format, and sends the results back to a (ii) **Query Processor**, which provides an interface to formulate SPARQL queries and distributes the queries among individual endpoints; furthermore, it retrieves the individual log graphs from the endpoints, integrates them, and presents the resulting integrated graph.

In the following, we explain the individual components in detail.

### SPARQL Query Editor

This Sub-Component is Part of the *Query Processor* and allows analysts to define settings for query execution, including: (i) *Target Hosts*: a set of endpoints to be included in the log analysis, (ii) *Knowledge bases*: a collection of internal and/or external sources of background knowledge that should be included in the query execution (e.g., IT infrastructure, cyber threat intelligence knowledge bases, etc.), (iii) *Time Interval*: the time range of interest for the log analysis (i.e., start time and end time).

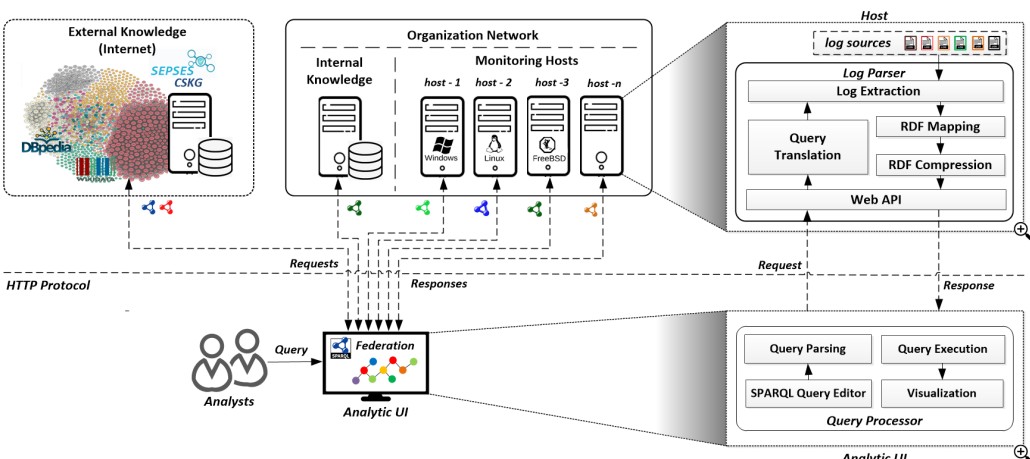

**Figure 4.** Virtual log graph and query federation architecture.

## Query Parsing

The SPARQL query specification [47] provides a number of alternative syntaxes to formulate queries. For uniform access to the properties and variables inside the query, we therefore parse the raw SPARQL syntax into a structured format prior to transferring the query to the monitoring hosts. The prepared SPARQL query is then sent as a parameter to the *Query Translator* via the *Web API* in the *Log Parser* Component.

## Query Translation

This sub-component decomposes the SPARQL query to identify relevant properties for log source selection and log line matching. Algorithm 1 outlines the general query translation procedure, which identifies relevant log sources and log lines based on three criteria, i.e., (i) prefixes used in the query; (ii) triples; and (iii) filters.

*Prefixes* $(P)$ is a set of log vocabulary prefixes that appear in a given query $Q$. In each query, the contained prefixes will be used by the query translator to identify relevant log sources. Available prefixes can be configured for the respective log sources in the *Log Parser* configuration on each client, including, e.g., the path to the local location of the log file. As an example, `PREFIX auth: <http://w3id.org/authLog>` is the prefix for *AuthLog*; its presence in a query indicates that the *AuthLog* on the selected hosts will be included in the log processing.

*Triples* $(T)$ is a set of triples that appear in a query, each represented as *Triple Pattern* or a *Basic Graph Pattern (BGP)* (i.e., `<Subject> <Predicate> <Object>`).

We match these triples to log lines (e.g., hosts and users) as follows: Function *getTriplePattern*$(Q)$ collects the triple patterns $T$ contained within the query $Q$. For each triple statement in a query, we identify the type of `Object` $T_{i_{Object}}$. If the type is *Literal*, we identify the $T_{i_{predicate}}$ as well. For example, for the triple {`?Subject cl:originatesFrom "Host1"`}, the function *getLogProperty*() identifies $T_{i_{Object}}$ "Host1", and additionally, looks up the property range provided in *regexPatterns* $(RP)$.

*regexPatterns* $(RP)$ links property terms in a vocabulary to the terms in a log entry and the respective regular expression pattern. For example, the property $cl : originatesFrom$ is linked to the concept `"hostname"` in *regexPattern* $(RP)$, which has a connected regex pattern for the extraction of host names. The output of the *getLogProperty*() function is a set of <*logProperty*, $T_{i_{Object}}$> key-value pairs.

Similar to triples, we also include *Filters* $(F)$ that appear in a query $Q$ for log-line matching. Filter statements contain the term `FILTER` and a set of pairs (i.e., *Variable* and *Value*), therefore each *Filter* statement $F_i$ has the members *Variable* $F_{i_{Variable}}$ and *Value* $F_{i_{Value}}$. Currently, we support FILTER clauses with simple pattern matching and regular expressions such as $FILTER\ (?variable = "StringValue")$, $FILTER\ regex(str(?variable), "StringValue"))$.

---

**Algorithm 1:** Query translation.

    size

    **Input:** SPARQL Query ($Q$), Vocabulary ($V$), regexPatterns ($RP$)

    **Output:** QueryElements ($Qe$)

1  Prefixes $P = \{P_1,...,P_n\} \in Q$ ;

2  Triples $T = \{Subject, Predicate, Object\} \in Q$ ;

3  Filters $F = \{Variable, Value\} \in Q$;

4  **Function** `translateQuery(`$Q,V,RP$`)`:

5     $P \leftarrow getPrefix(Q)$;

6     $T \leftarrow getTriplePattern(Q)$;

7     **foreach** *Triple $T_i \in T$* **do**

8       **if** $type(T_{i_{Object}})$=*Literal* **then**

9          $logProperty \leftarrow getLogProperty(T_{i_{Predicate}},V,RP)$;

10          $keyValue \leftarrow \{logProperty, T_{i_{Object}}\}$;

11       **end**

12       $triplesKV \mathrel{+}= keyValue$;

13     **end**

14     $F \leftarrow getFilterStatement(Q)$;

15     **foreach** *Filter $F_i \in F$* **do**

16       **if** $type(F_{i_{Value}})$=*Literal* **then**

17          $predicate \leftarrow getPredicate(Q,F_{i_{Variable}})$;

18          $logProperty \leftarrow getLogProperty(predicate,V,RP)$;

19          $keyValue \leftarrow \{logProperty, F_{i_{Value}}\}$;

20       **end**

21       $filtersKV \mathrel{+}= keyValue$;

22     **end**

23     $Qe \leftarrow \{P,triplesKV,filtersKV\}$;

24     **return** $Qe$;

25  **End Function**

---

The function *getFilterStatement*($Q$) is used to retrieve these filter statements from the query and to identify the type of *Value $F_{i_{Value}}$*. If it is a Literal, the *getPredicate*($Q$) function will look for the connected *predicate*. Similar to the technique used in triples, we use *getLogProperty*() to retrieve the regular expression defined in *regexPattern* ($RP$). Finally, the collected prefixes and retrieved key-value pairs, both from triples and filters, will be stored in *QueryElements* (*Qe*) for further processing. Figure 5 depicts a SPARQL query translation example.

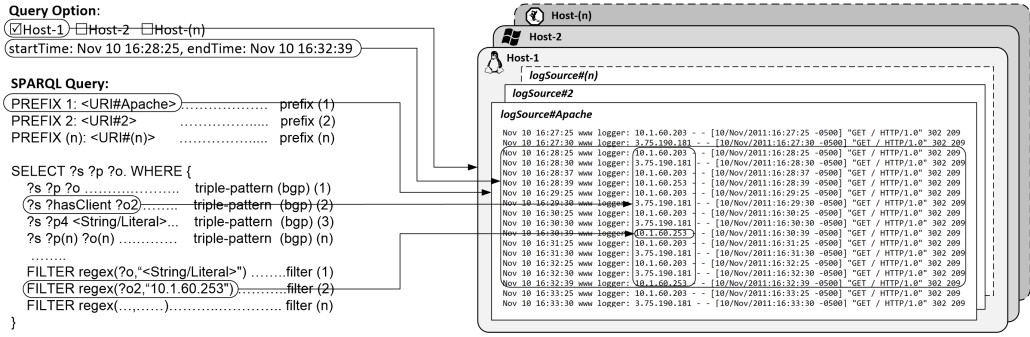

**Figure 5.** SPARQL Query translation example.

**Log Extraction** This component is part of the *Log Parser* that extracts the selected raw log lines and splits them into a key-value pair representation by means of predefined regular

expression patterns. As outlined in Algorithm 2, Log sources ($Ls$) are included based on the prefixes that appear in the query.

---

**Algorithm 2:** Log Extraction and RDF Mapping.

    size
    **Input:** SPARQL Query ($Q$), TimeFrame ($Tf$), LogSources ($Ls$)
    **Output:** Response ($R$)

1   TimeFrame $Tf = \{startT, endT\}$ ;
2   LogSources $Ls = \{Ls_1, ..., Ls_n\}$;
3   LogLines $Ln = \{Ln_1, ..., Ln_n\} \; \epsilon \; Ls$;
4   LogSourceOptions $LsO = \{vocabulary, regexPatterns\} \; \epsilon \; Ls$;
5   LogLineOptions $LnO = \{logTime, logProperties\} \; \epsilon \; Ln$ ;
6   QueryElements $Qe = \{prefixes, triplesKV, filtersKV\}$;
7   $Qe \leftarrow translateQuery(Q, LsO_{vocabulary}, LsO_{regexPatterns})$;
8   **foreach** *LogSource $Ls_i \; \epsilon \; Ls$* **do**
9     **if** $Qe_{prefixes}$ *contains* $LsO_{i_{vocabulary}}$ **then**
10       **foreach** *LogLines $Ln_j \; \epsilon \; Ln$* **do**
11         $lt \leftarrow LnO_{j_{LogTime}}$;
12         **if** $lt < Tf_{endT} = False$ **then**
13           break;
14         **end**
15         **if** $lt > Tf_{startT}$ && $lt < Tf_{endT}$ **then**
16           $ml \leftarrow matchLog(LnO_{j_{logProperties}}, Qe_{triplesKV}, Qe_{filtersKV})$;
17           **if** $ml = True$ **then**
18             $parsedLine \leftarrow parseLine(Ln_j)$;
19           **end**
20         **end**
21         $parsedData \mathrel{+}= parsedLine$;
22       **end**
23       $RDFData \leftarrow RDFMapping(parsedData)$;
24       $result \leftarrow compressData(RDFData)$;
25       **if** $result = True$ **then**
26         $response \leftarrow "Success"$;
27       **end**
28     **end**
29     **return** *response*;
30 **end**

---

For each log line ($Ln_j$) in a log source, we check whether the log timestamp ($LnO_{logTime}$) is within the defined TimeFrame ($Tf$). We leverage the monotonicity assumption that is common in the log context by stopping the log parsing once the end of the temporal window of interest is reached in a log file (i.e., we assume that log lines do not appear out of order). This can be adapted, if required for a specific log source. If this condition is satisfied, the *matchLog()* function checks the logline property ($LnO_{logProperties}$) against the set of queried triples ($Qe_{triplesKV}$) and filters ($Qe_{filtersKV}$). If the log line matches the requirements, the selected log line will be parsed using *parseLine()* based on predefined regular expression patterns. The resulting parsed queries will be accumulated and cached in a temporary file for subsequent processing.

### RDF Mapping

This sub-component of the *Log Parser* maps and parses the extracted log data into RDF. It uses the standard RDF mapping language to map between the log data and the vocabulary. Different log sources use a common core log vocabulary (e.g., SEPSES coreLog [48]) for

common terms (e.g., host, user, message) and can define extensions for specific terms (e.g., the `request` term in ApacheLog). The RDF Mapping also maps terms from a log entry to specific background knowledge (e.g., hosts in a log entry are linked to their host `type` according to the background knowledge). Figure 6 provides an overview of the log graph generation process.

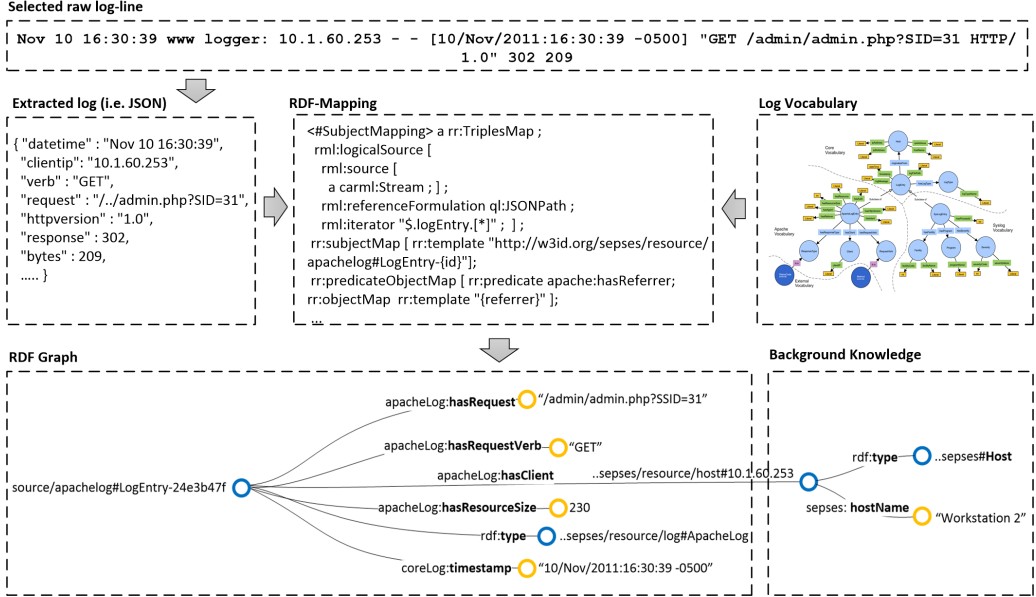

**Figure 6.** Log graph generation overview.

## RDF Compression

This sub-component is part of the *Log Parser*, which transforms the resulting RDF output produced by the *RDF Mapper* into a compact version of RDF. This compression results in a size reduction by an order of magnitude, which has significant advantages in our *VloGraph* framework: (i) it enables fast data transfer to the *Query Processor* component and thereby reduces latency; (ii) it makes the query execution itself more efficient as the compressed RDF version enables query operations without prior decompression directly on the binary representation [49].

We discuss the implementation of this component based on existing libraries in Section 6 and evaluate the effect of compression on the query execution performance on virtual log graphs in Section 7.

## Query Execution

Once the pre-processing on each target host has been completed and the compressed RDF data results have been successfully sent back to the *Query Processor*, a query engine executes the given queries against the compressed RDF data. If multiple hosts were defined in the query, the query engine will perform query federation over multiple compressed RDF data from those individual hosts and combine the query results into an integrated output.

Furthermore, due to semantic query federation, external data sources are automatically linked in the query results in case they were referenced in the query (cf. Section 6 for an example that links IDS messages to the SEPSES-CSKG [50]).

## Visualization

Finally, this component presents the query results to the user; depending on the SPARQL query form [51], e.g.,: (i) SELECT—returns the variables bound in the query pattern, (ii) CONSTRUCT—returns an RDF graph specified by a graph template, and (iii) ASK—returns a Boolean indicating whether a query pattern matches.

The returned result can be either in JSON or RDF format, and the resulting data can be presented to the user as an HTML table, chart, graph visualization, or it can be downloaded as a file.

## 6. Implementation & Application Scenarios

In this section, we discuss the implementation of *VloGraph* framework Source code available at Github and demonstrate its feasibility by means of three application scenarios.

### 6.1. Implementation

The *VloGraph* prototype relies on a number of existing open source tools and libraries. Specifically, we implement the *Log Parser* component as a Java-based tool that is installed and run on each monitoring host. It supports log parsing from multiple different OSs (e.g., Windows, Linux, etc.) and heterogeneous log files (e.g., authlog, apachelog, IISlog, IDSlog). For the *Log Extraction* component, we integrate *Grok Patterns*, a collection of composeable regular expression patterns that can be reused across log sources. Furthermore, we use *CARML* [52] as an *RDF Mapping* component based on RML mappings [53] to map the extracted log data into RDF. For the *RDF Compression* component, we leverage the *HDT* [49] library to efficiently compress the resulting RDF data into a compact, binary format that allows query operations without prior decompression.

For the analysis interface, we implemented a *Query Processor* component as a web-application that receives SPARQL queries, sends them to multiple target hosts, and presents the resulting graph to the analyst. Figure 7 shows the user interface of the application, which consist of (i) Query Options, including e.g., target hosts, background knowledge, analysis timeframe, as well as predefined queries to select. (ii) SPARQL Query Input to formulate and execute SPARQL queries, and (iii) Query Results to present the output of the executed query.

The query execution is implemented on top of the *Comunica* [54] query engine that supports query federation over multiple linked data interfaces including HDT files and SPARQL endpoints.

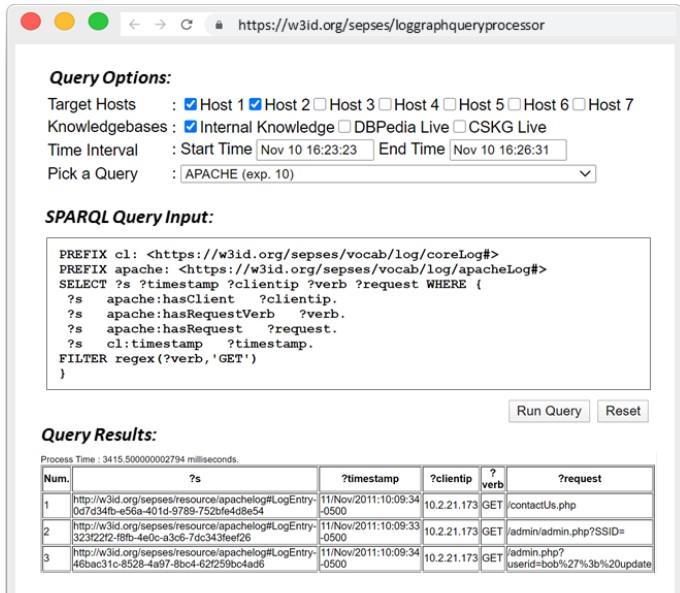

**Figure 7.** SPARQL query editor interface.

### 6.2. Application Scenarios

We demonstrate the feasibility of the *VloGraph* framework by means of three application scenarios i.e., (i) Web access log analysis; (ii) Network monitoring that demonstrates

the use of internal background knowledge; and (iii) Threat detection based on existing community rules and integration with the ATT&CK knowledge graph.

### Scenario I—Web Access Log Analysis

In this scenario, we simulated two hosts (Windows10 and Ubuntu) with different web servers (Apache and IIS) and analyze their access logs together. In order to identify access from a specific IP address (e.g., 192.168.2.1), we formulate the SPARQL query depicted in Listing 1. We specify the client's IP address with `access:hasClient res:ip-192.168.2.1` and filter for "GET" requests via `accs:hasRequestVerb res:GET`. In the query options, we selected the timeframe (from 11 November 2021 10:00:04 to 11 November 2021 10:10:04) as well as the two target hosts.

```
PREFIX cl: <https://w3id.org/sepses/vocab/log/coreLog#>
PREFIX accs: <https://w3id.org/sepses/vocab/log/accessLog#>
PREFIX res: <https://w3id.org/resource/access#>

SELECT ?logType ?hostOS ?hostIp  ?verb ?request
    WHERE {
        ?logEntry cl:originatesFrom ?host.
        ?host cl:hostOS ?hostOS.
        ?logEntry cl:hasLogType ?logType.
        ?host cl:ipAddress ?hostIp.
        ?logEntry accs:hasRequestVerb res:GET.
        ?logEntry accs:hasRequest ?request.
        ?logEntry accs:hasClient res:ip-192.168.2.1.
    } LIMIT 4
```

**Listing 1.** Web access query.

The query results in Table 1 show the access information with their log sources and types (`cl:IIS` and `cl:apache`), the host OS (`Win10` and `ubuntu`) with their IPs, the request method, and request paths. Figure 8 depicts the graph visualization of the result.

**Table 1.** Web access query result (excerpt).

| logType | hostOS | hostIp | Verb | Request |
|---------|--------|--------|------|---------|
| IIS | Win10 | 192.168.0.113 | *GET* | /employee.asp&id=12345 . . . |
| apache | Ubuntu | 192.168.0.111 | *GET* | /admin.php?userid=bob. . . |
| apache | Ubuntu | 192.168.0.111 | *GET* | /salary.php |
| IIS | Win10 | 192.168.0.113 | *GET* | /global/lwb.min.js . . . |

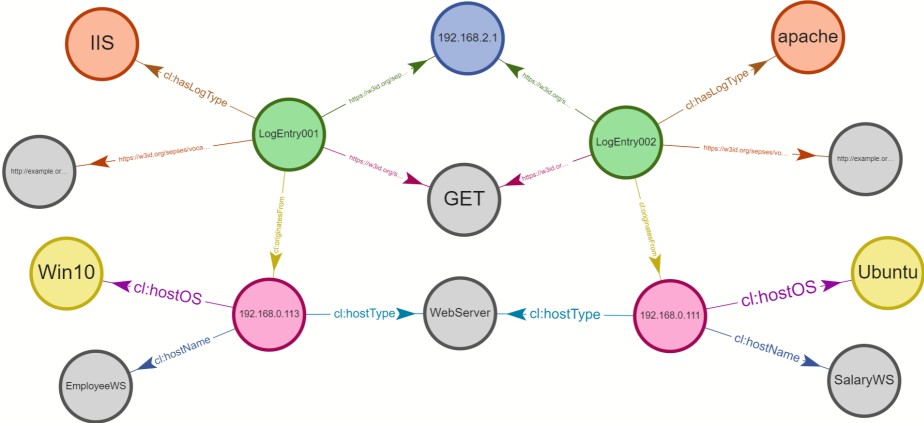

**Figure 8.** Web access query result visualization (excerpt).

**Scenario II—Network Monitoring**

In this scenario, we illustrate how our prototype provides semantic integration, generalization, and entity resolution. We simulated SSH login activities across different servers (e.g., *DatabaseServer, WebServer, FileServer*) with multiple demo users (e.g., *Bob and Alice*) and then queried the authlog files with our federated approach.

Typically, atomic information on the log entry level is not explicitly linked to semantic concepts. Hence, we added extractors to, e.g., detect specific log events from log messages and map them to event types from our internal background knowledge (e.g., `event:Login`, `event:Logout`). Furthermore, we added concept mappings for program names, IP addresses etc. (cf. Section 5).

Now, an analyst can formulate a SPARQL query as shown in Listing 2 to extract successful login events from SSH connections. The query results in Table 2 and Figure 9 show successful logins via SSH over multiple hosts in the specified time range (from 11 Decmeber 2021 13:30:23 to 11 Decmeber 2021 14:53:06). The host type and target IP address come from internal background knowledge, as the host name is connected to a specific host type.

This information can be a starting point for security analysts to explore the rich context of the events in the virtual knowledge graph.

**Table 2.** SSH connections query result (excerpt).

| Timestamp | User | sourceIp | targetHostType | targetIp |
|-----------|------|----------|----------------|----------|
| Dec 10 13:30:23 | Bob | 172.24.66.19 | DatabaseServer | 192.168.2.1 |
| Dec 10 13:33:31 | Alice | 172.24.2.1 | WebServer | 192.168.2.2 |
| Dec 10 13:38:16 | Alice | 172.24.2.1 | DatabaseServer | 192.168.1.3 |
| Dec 10 14:53:06 | Bob | 172.24.66.19 | FileServer | 192.168.2.4 |

```
PREFIX cl: <https://w3id.org/sepses/vocab/log/core#>
PREFIX auth: <https://w3id.org/sepses/vocab/log/authLog#>
PREFIX sys: <https://w3id.org/sepses/resource/system#>
PREFIX ev: <https://w3id.org/sepses/resource/event#>

SELECT ?timestamp ?user ?sourceIp ?targetHostType ?targetIp
    WHERE {
        ?logEntry cl:timestamp ?timestamp.
        ?logEntry auth:hasUser ?user.
        ?logEntry auth:hasSourceIp ?sourceIp.
        ?logEntry auth:hasTargetHost ?th.
        ?logEntry auth:hasAuthEvent ?ae.
        ?ae sys:partOfEvent ev:Login.
        ?th sys:hostType ?targetHostType.
        ?th cl:IpAddress ?targetIp.
    } LIMIT 4
```

**Listing 2.** SSH connections query.

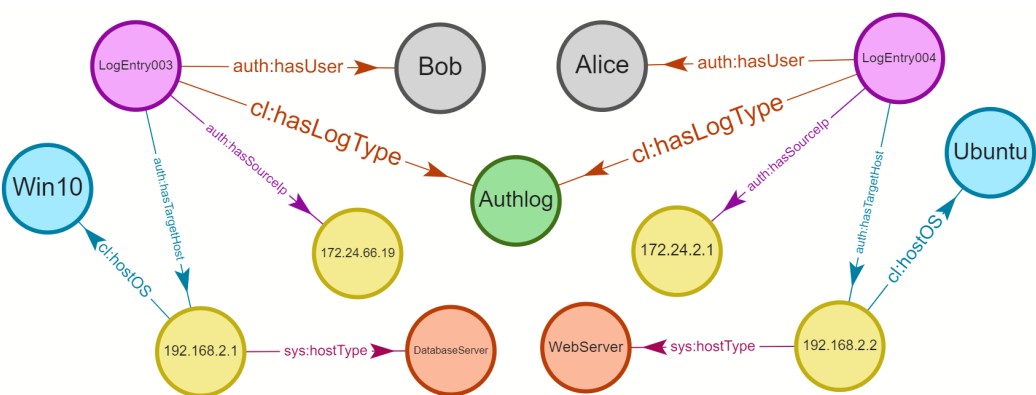

**Figure 9.** SSH connections query result visualization (excerpt).

**Scenario III—Threat Detection and ATT&CK Linking**

In this scenario, we demonstrate how the *VloGraph* framework leverages existing threat detection rules to identify Indicators of Compromise (IoCs) from log sources and link them to the respective attack techniques and tactics. For this scenario, we used an existing log dataset [6] as described in the motivation example in Section 1. To define our rule-based threat detection queries, we relied on existing community-based threat detection rules such as *Sigma* [55] and transformed them into RDF/SPARQL. Furthermore, we used the ATT&CK-KG [56], a continuously updated cybersecurity knowledge graph generated from the MITRE ATT&CK Matrix [10] in order to link cyber attacks to adversary techniques and tactics.

Listing 3 shows an example query for this scenario. Using the transformed *Sigma* rule as internal knowledge, we can list *suspicious* keywords defined in the rules (i.e., via `?sigma sigma:keywords ?keywords`) and use them to filter messages from the targeted log sources. In this case, we target request messages in Apache log (see `?logEntry apache:hasRequest ?req`) and filter them against the keywords (`FILTER regex(str(?req), ?keywords)`). Next, we link the detected log entries to the respective attack techniques (note that *Sigma* typically provides *tags* that associate its rules with ATT&CK techniques). This can be performed via `?sigma rule:hasAttackTechnique ?techn`. The query leverages linked data principles to include external background knowledge from the ATT&CK-KG, which makes it possible to further link the identified attack technique detailed knowledge such as technique description (via `?techn dcterm:description ?desc`), attack tactic (via `?techn attack:accomplishesTactic ?tactic`), CAPEC [57] attack patterns (`?techn attack:hasCAPEC ?capec`), and so forth.

Table 3 and Figure 10 show the query results and visualization from this scenario. Several log entries from a particular host (mail.cup) are associated with suspicious keywords. For example, according to a Sigma rule (Webshell Keyword), included as background knowledge, the "whoami" keyword is considered indicative of a *Web Shell* attack technique (T1505.003). This technique in turn is an instance of the tactic Persistence and of attack pattern CAPEC-650.

```
PREFIX cl: <https://w3id.org/sepses/vocab/log/core#>
PREFIX apache: <https://w3id.org/sepses/vocab/log/apache#>
PREFIX sigma: <http://w3id.org/sepses/vocab/rule/sigma#>
PREFIX rule: <http://w3id.org/sepses/vocab/rule#>
PREFIX attack: <http://w3id.org/sepses/vocab/ref/attack#>
PREFIX dcterm: <http://purl.org/dc/terms/>

SELECT ?logEntry  ?timestamp ?host ?keywords ?techn ?desc ?tactic ?capec
  WHERE {
    ?logEntry apache:hasRequest ?req ;
              cl:originatesFrom ?host;
              cl:timestamp ?timestamp.
    FILTER regex(str(?req),?keywords)
       { SELECT ?keywords ?techn ?tactic {
            ?sigma sigma:keywords ?keywords.
            OPTIONAL {
                ?sigma rule:hasAttackTechnique ?techn.
                ?techn dcterm:description ?desc.
                ?techn attack:accomplishesTactic ?tactic.
                ?techn attack:hasCAPEC ?capec.
            }
       }}
    } LIMIT 4
```

**Listing 3.** Rule-based threat detection and ATT&CK linking query.

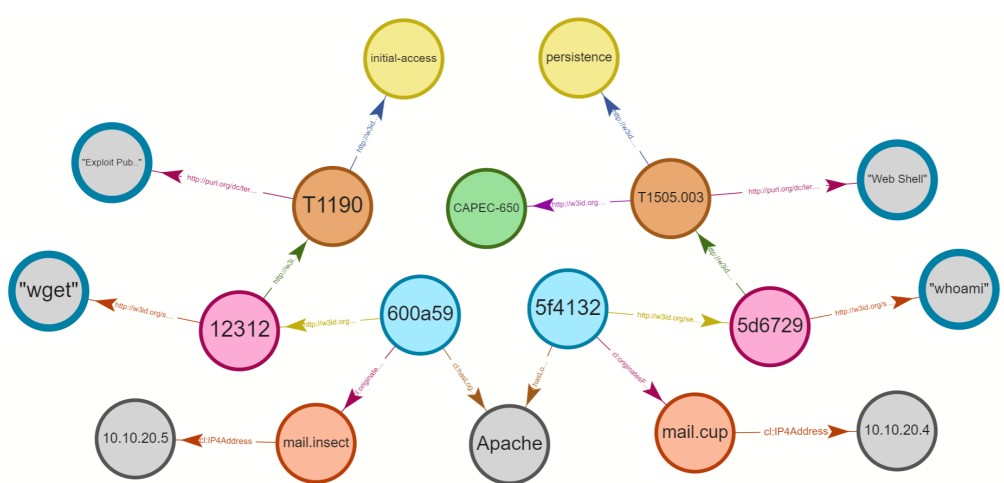

**Figure 10.** Threat detection and ATT&CK linking visualization (excerpt).

**Table 3.** Scenario 4 Query Results (Excerpt).

| logEntry | Timestamp | Host | Keywords | Techn | Desc | Tactic | Capec |
|----------|-----------|------|----------|-------|------|--------|-------|
| 5f4a32… | Mar 04 19:18:43 | cup | "whoami" | T1505.003 | *"Web Shell"* | persistence | CAPEC-650 |
| 468226… | Mar 04 14:05:41 | insect | "whoami" | T1505.003 | *"Web Shell"* | persistence | CAPEC-650 |
| 7cff1d1… | Mar 04 19:18:46 | cup | "curl" | T1190 | *"Exploit Pub.."* | initial-access | - |
| 600a59… | Mar 04 19:18:43 | insect | "wget" | T1190 | *"Exploit Pub.."* | initial-access | - |

## 7. Evaluation

We evaluated the scalability of our approach by means of a set of experiments in non-federated and federated settings.

### 7.1. Evaluation Setup

The experiments were carried out on Microsoft Azure virtual machines with seven hosts (4 Windows and 3 Linux) with 2.59 GHz vCPU and 16 GB RAM each. We reused the log vocabularies from [17] and mapped them to the log data.

**Dataset Overview**

We selected the systematically generated AIT log dataset (V1.1) that simulates six days of user access across multiple web servers including two attacks on the fifth day [6]. As summarized in Table 4, the dataset contains several log sources from four servers (*cup*, *insect*, *onion*, *spiral*).

To reduce reading overhead and improve log processing performance, we split large log files from the data set into smaller files—this can easily be replicated in a running system using log rotation mechanisms. Specifically, we split the files into chunks of 10k–100k log lines each and annotated them with original filename and time-range information

**Table 4.** Dataset description.

| LogType | #Properties | mail.cup.com | | mail.insect.com | | mail.onion.com | | mail.spiral.com | |
|---|---|---|---|---|---|---|---|---|---|
| | | Size | #Lines | Size | #Lines | Size | #Lines | Size | #Lines |
| Audit | 36 | 25 GB | 123.6 M | 22.7 GB | 99.9 M | 14.6 GB | 68.8 M | 12.4 GB | 59.5 M |
| Apache | 12 | 36.9 MB | 148 K | 44.4 MB | 169.3 K | 22.7 MB | 81.9 K | 24.8 MB | 100.4 K |
| Syslog | 6 | 28.5 MB | 158.6 K | 26.9 MB | 150.7 K | 15 MB | 86.6 K | 15.1 MB | 85.5 K |
| Exim | 11 | 649 KB | 7.3 K | 567 KB | 6.2 K | 341 KB | 3.9 K | 355 KB | 4 K |
| Authlog | 11 | 128 KB | 1.2 K | 115 KB | 1.1 K | 102 KB | 1 K | 127 KB | 1.2 K |

### 7.2. Single-Host Evaluation

We measured the overall time for virtual log graph processing including (i) log reading (i.e., searching individual log lines), (ii) log extraction (i.e., extracting the raw log line into structured data), (iii) RDF Mapping (i.e., transforming json data into RDF), and (iv) RDF compression (i.e., compressing RDF into Header, Dictionary, Triples (HDT) format).

In our scenarios, we included several log sources; for each log source, we formulated a SPARQL query to extract 1k, 3k, 5k, and 7k log lines filtering by timestamp in the query option. We report the average times over five runs for experiments with several log sources—i.e., Auditlog (AD), Apache for web logs (AP), Exim for mail transfer agent logs (EX), Syslog for Linux system logs (SY), and Authlog for authentication logs (AT)—for a single host in Figure 11. We used the data set from the first web server (i.e., mail.cup.com) in this evaluation. Note that we only extracted 1000k log lines from Authlog due to the small original file size (less than 1.2k log lines).

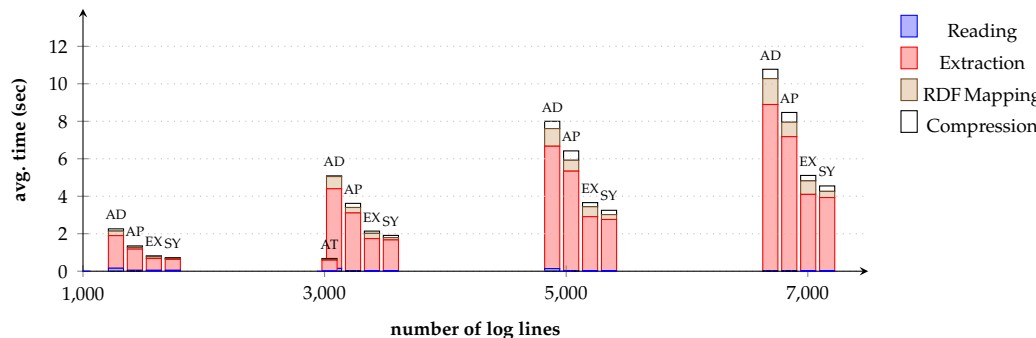

**Figure 11.** Average log graph generation time for *n* log lines with a single host (36 extracted properties).

We found that the performance for log graph extraction differs across the log sources. Constructing a log graph from Auditlog (AD) data resulted in the longest processing times followed by Apache, Exim, Syslog and Authlog. The overall log processing time scales linearly with the number of extracted log lines. Typically, the log extraction phase accounts for the largest proportion (>80%) of the overall log processing time. Furthermore, we found that the increase in log processing time with a growing number of extracted log lines is moderate, which suggests that the approach scales well to a large number of log lines.

**Dynamic Log Graph Generation**

As discussed in the first part of the evaluation, execution times are mainly a function of the length of text in the log source and the granularity of the extraction patterns (i.e., log properties). As can be seen in Table 4, the log sources are heterogeneous and exhibit different levels of complexity. In our setup, Auditlog, for instance, has the largest number of log properties (36), followed by Apache (12), Exim (11), Authlog (11), and Syslog (6).

Figure 12 shows an evaluation of log graph generation performance with respect to the complexity of the log source. We use the *Auditlog* for this evaluation as it has the highest number of log properties. Overall, the log graph generation performance grows linearly with the number of extracted log properties. Hence, queries that involve a smaller subset of properties (e.g., only *user* or *IP address* rather than all information that could potentially be extracted) will typically have smaller generation times.

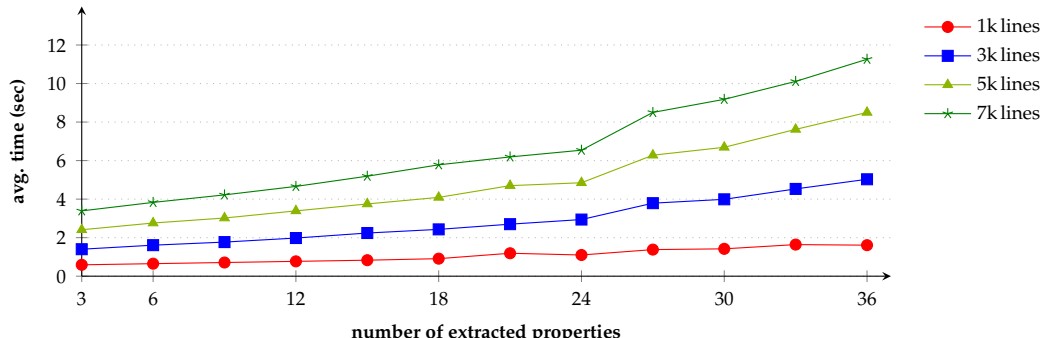

**Figure 12.** Dynamic log graph generation time. Experiments carried out on AuditLog data on a single host.

**Graph Compression**

Figure 13 shows the performance for log graph compression on the *Auditlog* dataset.

We performed full property extraction (i.e., all 36 identified properties) against 5k, 10k, 15k, and 20k log-lines, respectively, and compare the original size of raw log data, the generated RDF graph in TURTLE [19] format (.ttl), and the compressed graph output in the HDT format.

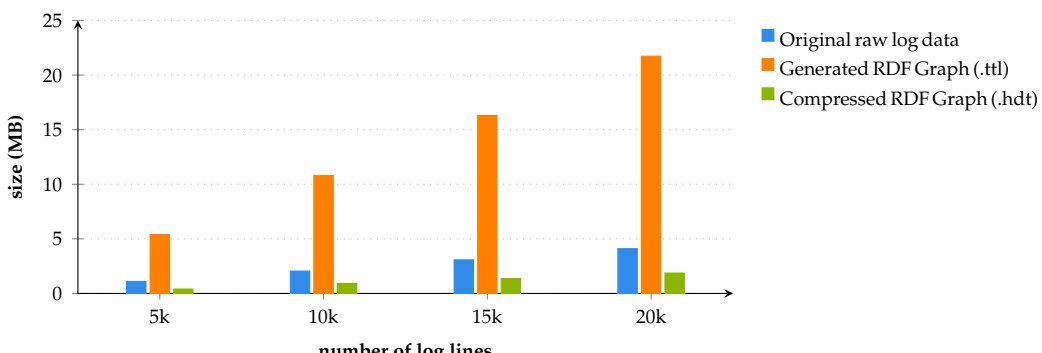

**Figure 13.** Graph compression.

For 5k log lines (1 MB raw log) compression results in approximately 0.4 MB compared to 5.4 MB for the uncompressed RDF graph. 20k log lines (4 MB raw log) compresses to about 1.87 MB from 21.4 MB uncompressed generated RDF graph. Overall, the compressed version is typically less than half the size of the original raw log and 10 times smaller than the generated RDF graph. The resulting graph output would be even smaller for fewer extracted properties, minimizing resource requirements (i.e., storage/disk space).

*7.3. Multi-Host Evaluation*

To evaluate the scalability of our approach, we measured the log processing time for multiple hosts on the same network. This evaluation includes not only the log processing but also the query federation performance. Federation means that the queries are not only executed concurrently, but that they involve evaluating and combining individual query results to achieve integrated results.

Table 5 summarizes the evaluation setup that consists of six experiments ranging from 30 min up to 5 h. The timeframe describes the starting time and the end time of analysis; log lines per host summarizes the range of log lines per host within the timeframe. For this evaluation, we used the Apache log dataset described in Table 4 and conducted the analysis within the log timeframe of 2 March 2020, starting from 8 pm. Host 1 to host 4 store the data from the original four servers in the dataset (host 1 mail.cup.com, host 2 mail.insect.com, and so on); for the three additional hosts in the evaluation, we replicated the log files from mail.cup.com, mail.insect.com, and mail.spiral.com. Similar to the single-host evaluation, for each experiment, we reported the average times over five runs.

**Table 5.** Multihost Experiment Timeframe.

| Experiment | Duration | Log Lines per Host | Experiment | Duration | Log Lines per Host |
|---|---|---|---|---|---|
| E1 | 30 min | 0.7k–1k | E4 | 3 h | 3k–5k |
| E2 | 1 h | 1k–1.7k | E5 | 4 h | 6k–8k |
| E3 | 2 h | 2.8k–4k | E6 | 5 h | 8k–10k |

Figure 14 shows the average log processing times for each experiment. The 1 h experiment shows that log processing for two hosts takes approx. 4.7 s on average. In the same experiment, the time slightly increases with an increasing number of hosts and reaches a max. of 7.5 s. The log processing time for the 5 h experiment with two hosts takes approximately 19.01 s on average and reaches the maximum average time of 26.10 s with 7 hosts. Based on these results, we conclude that the growth of the log processing time as a function of the number of hosts is moderate. Therefore, this approach scales well with a growing number of hosts to monitor, as the log processing on each host is parallelized and the query federation overhead is low.

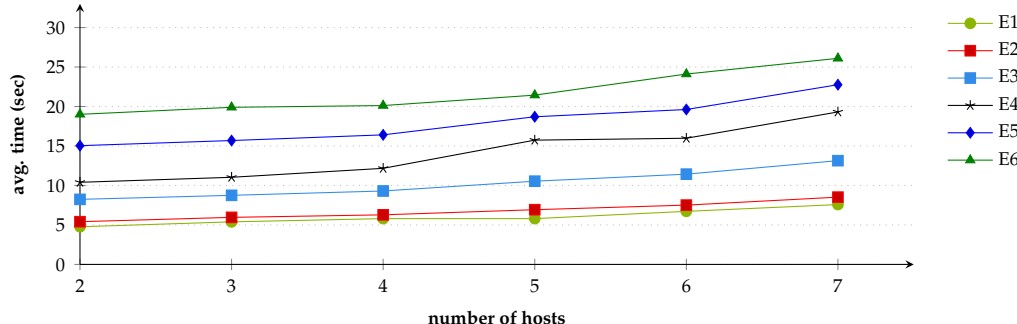

**Figure 14.** Query execution time in a federated setting for different time frames. Evaluation of linking to background knowledge stored on external servers is out of scope.

## 8. Discussion

In this section, we reflect upon benefits, limitations, and possible applications of the proposed virtual log knowledge graph framework.

*Graph-Based Log Integration and Analysis*

Representing log data in semantic graph structures opens up new possibilities, such as handling log data in a uniform representation, exploring connections between disparate entities, and applying graph-based queries to search for abstract or concrete patterns of log events. Compared to text-based search, graph-pattern based queries are more expressive and make it possible to link entities that appear in log lines to background knowledge. Furthermore, the ability to provide query results as a graph enables new workflows for analysts and may help them to be more efficient in exploring log data and ultimately improving their situational awareness faster.

In our evaluation, we find that SPARQL as a standardized RDF query language provides powerful means for graph pattern-based ad-hoc log analyses. A challenge, however, is that analysts are typically not familiar with the language and require some training. This may improve in the future, as SPARQL is often already part of computer science curricula and is increasingly being adopted in many industries [58]. Furthermore, intuitive general-purpose visual query building and exploration tools such as [59,60] could be used and possibly adapted for security applications to abstract the complexity of writing queries directly in SPARQL.

*Decentralization and Virtualization*

Decentralized ad-hoc extraction on the endpoints at query execution time is a particularly useful approach in scenarios where log acquisition, aggregation, and storage are difficult or impractical. This includes scenarios with a large number of distributed hosts and log sources. Pushing log analysis towards the endpoints is also particularly interesting in settings where bandwidth constraints do not permit continuous transmission of log streams to a central log archive.

Whereas these considerations apply generally, the decentralized approach also has benefits that are specific to our knowledge-graph based approach for log integration and analysis. Specifically, the federated execution distributes the computational load of extraction, transformation, and (partly) query execution towards the endpoints. This will be useful in many practical settings where the scale of the log data that is constantly generated in a distributed environment is prohibitively large and it is not feasible to transform the complete log data into a Knowledge Graph (KG) presentation. In such settings, the decentralized approach facilitates ad-hoc graph-based analyses without the need to set up, configure and maintain sophisticated log aggregation systems.

Our evaluation showed that this ad-hoc extraction, transformation, and federated query execution works efficiently for temporally restricted queries over dispersed log data without prior aggregation and centralized storage. Consequently, the approach is particularly useful for iterative investigations over smaller subsets of distributed log data that start from initial indicators of interest. It supports diagnostics, root cause analyses etc. and can leverage semantic connections in the graph that would otherwise make manual exploration tedious. An inherent limitation, however, is that the computational costs become exceedingly large for queries without any temporal restrictions or property-based filters—i.e., the approach is less useful for large-scale exploratory queries over long time intervals without any initial starting point.

*Log Parsing and Extraction*

The identification and mapping of relevant concepts in log messages is currently based on regular expression patterns. Extracted log lines are filtered and only lines that potentially match the query are transferred from the local endpoint, which minimizes bandwidth usage and processing load at the querying client. A limitation of this approach is that for complex

queries, the execution of a large set of regular expression patterns on each log line raises scalability issues.

An approach based on templates, similar to [16], could be applied to learn the structure and content of common log messages and then only extract the expected elements from those log messages. Furthermore, repeated application of regular expression patterns on each log line could also be avoided by building a local index on each endpoint. Such techniques should improve query performance, but these improvements have to be traded off against the additional complexity and storage requirements they introduce.

### Applications and Limitations

The illustrative scenarios in Section 6 highlighted the applicability of the approach in web access log analysis, intrusion detection, network monitoring, and threat detection and ATT&CK linking.

In these settings, ad-hoc integration of dispersed heterogeneous log data and graph-based integration can be highly beneficial to connect isolated indicators. Moreover, we found that the virtual log knowledge graph is highly useful in diagnostic applications such as troubleshooting or service management more generally and we are currently working on a framework for instrumenting containers with virtual knowledge graph interfaces to support such scenarios.

In the security domain—the focus in this paper—we found that virtual knowledge graphs can complement existing log analytic tools in order to quickly gain visibility in response to security alerts or to support security analysts in threat hunting based on an initial set of indicators or hypotheses.

Key limitations, however, include that the virtual integration approach is not directly applicable for (i) repeated routine analyses over large amounts of log data, i.e., in scenarios where up-front materialization into a KG is feasible and amortizes due to repeated queries over the same large data set or; (ii) continuous monitoring applications, i.e., scenarios where log data has to be processed in a streaming manner, particularly in the context of low latency requirements.

The latter would require the extension of the approach to streaming settings, which we plan to address in future work.

### Evasion and Log Retention

A typical motivation for shipping log data to dedicated central servers is to reduce the risk of undetected log tampering when hosts in the network are compromised. This reduces the attack surface, but makes securing the central log archive against tampering all the more critical. Relying on data extracted at the endpoints, by contrast, comes with the risk of local log tampering. File integrity features could help to spot manipulations of log files, but for auditing purposes, the proposed approach has to be complemented with secure log retention policies and mechanisms. Finally, the communication channel between the query processor in the analytic user interface and the local log parsers also represents an attack vector that has to be secured.

## 9. Conclusions

In this article, we presented *VloGraph*, a novel approach for distributed ad-hoc log analysis. It extends the Virtual Knowledge Graph (VKG) concept and provides integrated access to (partly) unstructured log data. In particular, we proposed a federated method to dynamically extract, semantically lift and link named entities directly from raw log files. In contrast to traditional approaches, this method only transforms the information that is relevant for a given query, instead of processing all log data centrally in advance. Thereby, it avoids scalability issues associated with the central processing of large amounts of rarely accessed log data.

To explore the feasibility of this approach, we developed a prototype and demonstrated its application in three log analysis tasks in security analytics. These scenarios demonstrate federated queries over multiple log sources across different systems. Fur-

thermore, they highlight the use of semantic concepts inside queries and the possibility of linking contextual information from background knowledge. We also conducted a performance evaluation which indicates that the total log processing time is primarily a function of the number of extracted (relevant) log lines and queried hosts, rather than the size of the raw log files. Our prototypical implementation of the approach provides scalability when facing larger log files and an increasing number of monitoring hosts.

Although this distributed ad-hoc querying has multiple advantages, we also discussed a number of limitations. First, log files are always parsed on demand in our prototype. By introducing a template-based approach to learn the structure of common log messages and by building an index on each endpoint to store the results of already parsed messages, query performance could be improved. Second, the knowledge-based ad-hoc analysis approach presented in this article is intended to complement, but does not replace traditional log processing techniques. Finally, while out of scope for the proof of concept implementation, the deployment of the concept in real environments requires traditional software security measures such as vulnerability testing, authentication, secure communication channels, and so forth.

In future work, we plan to improve the query analysis, e.g., to automatically select relevant target hosts based on the query and asset background knowledge. Furthermore, we will explore the ability to incrementally build larger knowledge graphs based on a series of consecutive queries in a step-by-step process. Finally, an interesting direction for research that would significantly extend the scope of potential use cases is a streaming mode that could execute continuous queries, e.g., for monitoring and alerting purposes. We plan to investigate this aspect and integrate and evaluate stream processing engines in this context.

**Author Contributions:** K.K.: Conceptualization, Methodology, Software, Investigation, Validation, Visualization, Writing—Original draft preparation. A.E.: Conceptualization, Writing—Review & Editing. E.K.: Conceptualization, Writing—Review & Editing. D.W.: Supervision. G.Q.: Supervision. A.M.T.: Supervision. All authors have read and agreed to the published version of the manuscript.

**Funding:** This research was funded by Netidee SCIENCE and Open Access Funding by the Austrian Science Fund (FWF) under grant P30437-N31. The competence center SBA Research (SBA-K1) is funded within the framework of COMET—Competence Centers for Excellent Technologies by BMVIT, BMDW, and the federal state of Vienna, managed by the FFG. Moreover, the financial support by the Christian Doppler Research Association, the Austrian Federal Ministry for Digital and Economic Affairs and the National Foundation for Research, Technology and Development is gratefully acknowledged (Christian-Doppler-Laboratory for Security and Quality Improvement in the Production System Lifecycle).

**Institutional Review Board Statement:** Not applicable.

**Informed Consent Statement:** Not applicable.

**Data Availability Statement:** The prototype and scenario data presented in this article are openly available on GitHub https://github.com/sepses/VloGParser (accessed on 24 February 2022). In the evaluation we also use the publicly available AIT Log Data Set V1.1 from Zenodo [10.5281/zenodo.4264796].

**Conflicts of Interest:** The authors declare no conflict of interest.

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
