# Peer review of "VloGraph: A Virtual Knowledge Graph Framework for Distributed Security Log Analysis"

_make, doi:10.3390/make4020016_

Round 1

Reviewer 1 Report

This work proposes a distributed log security analysis framework built on top of a virtual knowledge graph (represented in an RDF format) and a query language (SPARQL) that supports federated query analysis. The framework allows efficient querying of security logs of different sources and types. Given a query entry, the approach dynamically collects the relevant data from different sources and forms a knowledge graph that supports log analysis on top of it. The paper is well written and easy to follow. The approach is interesting and demonstrates practical benefits. Overall, it was a pleasant experience reading the paper. Below, I have some minor comments that may help the authors improve their work.

- In the Background section, the paper can illustrate a basic log knowledge graph to help readers better understand the content in the related work section. Otherwise, the discussions in the related work section may be too abstract for readers who are not familiar with log knowledge graphs.

- In the Background section, it would also be interesting to discuss the format and characteristics of security-related logs.

- In the evaluation section, in addition to the evaluation of the efficiency, the paper should also discuss the accuracy of the query results.

- What are the accuracy of different components (e.g., log parsing accuracy, complementness of log vocabulary) and how do they impact the query results?

- Can the approach be extended to support keyword or regex search of log lines?

- The paper should discuss its main extensions over the previously published conference version.

- Line#152-154: “To tackle these challenges, [17] create a foundation for semantic 152 SIEMs that introduces a Security Strategy Meta-Modelto enable interrelating information 153 from different domains and abstraction levels.” There should be a space between “Model” and “to”?

Author Response

Point 1: In the Background section, the paper can illustrate a basic log knowledge graph to help readers better understand the content in the related work section. Otherwise, the discussions in the related work section may be too abstract for readers who are not familiar with log knowledge graphs.

Response: Thank you for the recommendation - we added a basic log knowledge graph to the related work section.

Point 2: In the Background section, it would also be interesting to discuss the format and characteristics of security-related logs.

Response: Thank you for the recommendation, we extended the background section accordingly with a new subsection on log data and formats.

Point 3: In the evaluation section, in addition to the evaluation of the efficiency, the paper should also discuss the accuracy of the query results.

Point 4: What are the accuracy of different components (e.g., log parsing accuracy, complementness of log vocabulary) and how do they impact the query results?

Response to Point  4 & 5: Within the scope of the present paper, the focus is on ad-hoc extraction, query translation and federated execution, integration, and graph-based querying of virtual log knowledge graphs. For the first two illustrative scenarios presented in the paper, the relevance and completeness of the results is solely dependent on the correctness of the mappings and extraction patterns and on the query formulation. 

For scenarios that involve linking to background knowledge, such as the third scenario, the quality of the modeled knowledge (such as the Sigma rules in Scenario 3) will also affect the completeness and accuracy of the results. An evaluation of mappings and rules in specific scenarios is, however, beyond the scope of the present paper.

Point 6: Can the approach be extended to support keyword or regex search of log lines?

Response: Yes, SPARQL natively supports regex-based filter expressions on literals that can be used to filter based on the properties of returned log events.

Point 7: The paper should discuss its main extensions over the previously published conference version.

Response: We added a discussion on the extensions w.r.t. The previously published conference version to the introduction.

Point 7: Line#152-154: “To tackle these challenges, [17] create a foundation for semantic 152 SIEMs that introduces a Security Strategy Meta-Modelto enable interrelating information 153 from different domains and abstraction levels.” There should be a space between “Model” and “to”?

Response: Thank you, we have fixed it.

Reviewer 2 Report

This paper presents 'VloGraph: A Virtual Knowledge Graph Framework for
Distributed Security Log Analysis'. The idea is good but I have a few concerns:

  1. Please avoid 'we' if possible. Extensive use of 'we'.
  2. What is the new contribution in this paper when compared to REF 1?
  3. The conclusion part is sloppy. Please manage it in a good way. Please merge paragraphs. 
  4. The authors' contribution section is missing. 
  5. Few diagrams are not well explained. Please double-check this. 

Author Response

Point 1: What is the new contribution in this paper when compared to REF 1?

Response: We added a discussion on the extensions w.r.t. the previously published conference version to the introduction.

Point 2: The conclusion part is sloppy. Please manage it in a good way. Please merge paragraphs. 

Response: We condensed the conclusions and merged the paragraphs.

Point 3: The authors' contribution section is missing.

Response: Thank you for noting the omission - we added the authors' contribution section in the manuscript.

Point 4: Few diagrams are not well explained. Please double-check this. 

Response: We rechecked the captions of all figures and ensured that all of them are referenced and explained in the text.

Reviewer 3 Report

The paper presents a framework, named VloGraph, to help practitioners to visualize the relationships among log files in a graphical way, using a graph representation. 

The need for such type of framework is timely and well understood by reliability and security experts.

The paper extensively verifies the tool in the context of case studies and experiments also concerning the overhead and scalability.

The relation with related work is good, however some references are missing, for example on log-based error propagation analysis, which also makes use of graphs. Examples are: "Modeling soft-error propagation in programs", by Li et al, and "An empirical analysis of error propagation in critical software systems", by Cinque et al.

I see that the paper is an extended version of paper reference [1]. I believe authors should clearly state this in the introduction and clarify what are the differences between this paper and the earlier version.

Author Response

Point 1: The relation with related work is good, however some references are missing, for example on log-based error propagation analysis, which also makes use of graphs. Examples are: "Modeling soft-error propagation in programs", by Li et al, and "An empirical analysis of error propagation in critical software systems", by Cinque et al.

Response: Thank you for these very interesting pointers. Relating the proposed approach to the field of error propagation in software systems could be an interesting application. Within the scope of the present paper, however, we did not target this domain and generally did not include a full survey of potential other application domains due to space constraints.

Point 2: I see that the paper is an extended version of paper reference [1]. I believe authors should clearly state this in the introduction and clarify what are the differences between this paper and the earlier version.

Response: We added a discussion on the extensions w.r.t. The previously published conference version to the introduction.